# GPER Agonist G1 Prevents Wnt-Induced *JUN* Upregulation in HT29 Colorectal Cancer Cells

**DOI:** 10.3390/ijms232012581

**Published:** 2022-10-20

**Authors:** Maria Abancens, Brian J. Harvey, Jean McBryan

**Affiliations:** 1Department of Surgery, RCSI University of Medicine and Health Sciences, ERC Beaumont Hospital, D09 YD60 Dublin, Ireland; 2Department of Molecular Medicine, RCSI University of Medicine and Health Sciences, ERC Beaumont Hospital, D09 YD60 Dublin, Ireland; 3Department of Biology, Centro de Estudios e Cientificos CECs, Valdivia 5110466, Chile

**Keywords:** GPER, colorectal cancer, Wnt, beta-catenin

## Abstract

Women consistently show lower incidence and mortality rates for colorectal cancer (CRC) compared to men. Epidemiological evidence supports a pivotal role for estrogen in protecting women against CRC. Estrogen protective effects in CRC have been mainly attributed to the estrogen receptor beta (ERβ) however its expression is lost during CRC progression. The role of the G-protein coupled membrane estrogen receptor (GPER/GPER1/GPR30), which remains expressed after ERβ loss in CRC, is currently under debate. We hypothesise that estrogen can protect against CRC progression via GPER by modulating the Wnt/β-catenin proliferative pathway which is commonly hyperactivated in CRC. We sought evidence of sexual dimorphism within the Wnt/β-catenin pathway by conducting Kaplan–Meier analyses based on gene expression of the Wnt receptor *FZD1* (Frizzled 1) in multiple public domain CRC patient data sets. High expression of *FZD1* was associated with poor relapse-free survival rates in the male but not the female population. In female-derived HT29 CRC cell lines, we show that β-catenin nuclear translocation was not affected by treatment with the GPER agonist G1. However, G1 prevented the Wnt pathway-induced upregulation of the *JUN* oncogene. These novel findings indicate a mechanistic role for GPER in protecting against CRC progression by selectively reducing the tumorigenic effects of hyperactive Wnt/β-catenin signalling pathways in CRC.

## 1. Introduction

Colorectal cancer shows a clear sexual dimorphism worldwide with women presenting lower incidence (16.2 per 100,000 person-year) and mortality rates (7.2 per 100,000 person-year) compared to age-matched men (23.4 per 100,000 person-year; 11.0 per 100,000 person-year respectively) [1,2,3]. Evidence of a female protective effect against CRC is strong with sexual dimorphism consistently seen across continents and over multiple decades [4]. Endogenous female sex hormones are likely to play a part, though gender-associated differences in lifestyle factors, the use of screening, stage at diagnosis and site distribution of CRC may all contribute to survival differences between the sexes [4]. A number of studies have reported that the survival advantage of women over men is stronger at younger ages and reduced or lost as women pass the average age for natural menopause [5,6,7]. This distinction between pre- and post-menopausal years, reinforces the likelihood that estrogen plays a protective role in CRC.

There is mounting evidence, from population-based studies to animal and in vitro research, that estrogen can protect against CRC. For instance, Hormone Replacement Therapy has been associated with reduced CRC risk and mortality in postmenopausal women, with this effect being dependent on the starting point, duration or recentness of the treatment [8]. Similarly, oral contraceptive use is associated with lower risk of colorectal cancer [9]. The strongest preclinical data supporting an anti-tumour role for estrogen in CRC focuses on the nuclear estrogen receptor, ERβ. Both the AOM/DSS (Azoxymethane/Dextran Sodium Sulfate) and APC^min/+^ (Adenomatous Polyposis Coli) mouse models of CRC exhibit increased polyps/tumours when ERβ is knocked out [10,11]. Consistent with this, ERβ has been shown to inhibit cell proliferation in vitro, promote apoptosis and produce anti-inflammatory responses in CRC models [12,13,14]. However, loss of ERβ is a well-documented early event associated with CRC development and progression. This begs the question, can estrogen continue to produce protective effects and survival benefits after tumorigenesis following the loss of ERβ?

The membrane estrogen receptor GPER (also known as GPER1 or GPR30) is expressed in colonic epithelia and is the dominant estrogen receptor present once the tumour has developed [15]. Any potential effect of estrogens in the developed tumour would likely transduce via GPER. At the moment, our understanding of the role of GPER in CRC is in its infancy. One prominent study reported a tumour suppressor role for GPER and demonstrated that the GPER-specific agonist, G1, can reduce proliferation and enhance apoptosis in CRC cell lines as well as reducing tumour growth in a CRC tumour xenograft [16]. However, in the same year, another important study demonstrated that G1 can enhance proliferation in CRC cell lines. This effect was blocked by the GPER antagonist, G15, which reduced tumour growth in a CRC xenograft model, implying a pro-tumorigenic role for GPER [15]. One explanation put forward was that the experimental conditions and drug doses used in the two different studies may have had an impact, with GPER activation producing a biphasic response depending on the dose of ligand present. Indeed, research from our own group has identified that GPER activation can impact CRC cell proliferation and migration in an opposing manner depending on the level of oxygen present in the environment [17]. In fact, even the association between tumour GPER expression and patient survival is not yet clearly established. High expression of GPER protein expression has been correlated with good survival in a cohort of 90 patients [16] while high expression of GPER mRNA levels inversely correlated with survival in a cohort of 440 patients from the TCGA dataset [15]. An inverse correlation was also seen at the mRNA level in an analysis of 566 patients but this correlation was only detected in females with high stage tumours and not in females with low stage disease or males at any stage [17]. Clarifying the role of GPER by reproducing previously published results in independent settings and building on our knowledge of GPER activity is urgently needed if we are to improve our understanding of sexual dimorphism in colorectal cancer.

It is widely accepted that Wnt overactivation is the most relevant event of tumorigenesis in a large number of CRC cases [18]. Certainly, mutations in Wnt pathway members such as *APC* or *CTNNB1* that associate with increased activity levels of the pathway are found in over 90% of colorectal tumours [19,20]. Little attention has been paid to sexually dimorphic mechanisms of the Wnt pathway in CRC. Given the protective effects of estrogens in CRC, we hypothesize that estrogen through GPER activation may restrict Wnt activity in CRC cells to contribute to the female advantage in CRC. In this study, we show a potential relationship between the Wnt/β-catenin pathway and the estrogen receptor GPER in CRC.

## 2. Results

### 2.1. Wnt Receptor FZD1 Associates with CRC Patient Survival in a Sex-Specific Manner

To test our hypothesis, we first investigated if there was any evidence for sexual dimorphism within the Wnt signalling pathway in tumour tissue from colorectal cancer patients. To do this, we explored the association between gene expression of Frizzled1, a Wnt receptor, and survival in online publicly available patient datasets. The Guinney dataset is a compilation of gene expression data from >3000 CRC patient samples. Expert care was taken with pre-processing and normalisation to enable datasets from different sources and platforms to be merged for processing [21]. Using the R2 bioinformatics tool, Kaplan–Meier curves were plotted using gene expression data from the 802 patients with available survival information. High expression of *FZD1* significantly associated with poor relapse free survival in the male population (*n* = 439, median cut-off, *p* = 8.6 × 10^−4^). In the female cohort, no association between *FZD1* expression and survival was detected (*n* = 363, *p* = 0.187) (Figure 1). This analysis was repeated with the next 3 largest CRC datasets available in R2 bioinformatics tool: Marisa (*n* = 566), SieberSmith (*n* = 355) and Beissbarth (*n* = 363) [22,23]. For all 3 datasets, separation between the tracks was seen in the male population. Male patients with high *FZD1* expression displayed worse relapse-free survival than low expressing male patients. This separation reached significance in the Marisa (*p* = 0.015) and SiebersSmith (*p* = 0.021) datasets, but not the Beissbarth dataset which has considerably shorter follow-up time (*p* = 0.161). By contrast, no association between *FZD1* expression and relapse-free survival was detected in the female population in any of the datasets (Figure 1). Thus, the association of *FZD1* gene expression with CRC patient survival occurs in a sex-specific manner.

### 2.2. GPER Is the Predominant Estrogen Receptor Expressed in Colorectal Cancer Cell Lines

ERβ rather than ERα is the primary estrogen receptor expressed in healthy colonic epithelium. However, it is well recognised that ERβ expression is lost as colorectal cancer develops. Indeed, maintaining ERβ expression is one of the foremost mechanisms proposed for how estrogen can exert a protective effect in CRC. A mechanism for how estrogen may exert an effect after ERβ loss remains under discussion. We began by profiling the expression of ERα, ERβ and G-protein coupled estrogen receptor (GPER) in a panel of human colorectal cancer cell lines (HT29, DLD1, SW620 and T84 cells). ERα was undetectable at the protein level and beyond the level of reliable detection by real-time PCR (C_T_ > 35) in all 4 CRC cell lines (Figure 2). ERβ was detectable but at very low levels in the CRC cell lines. By contrast, GPER was strongly expressed at mRNA and protein levels in all 4 CRC cell lines with C_T_ values in the range of 23–29. For comparison, protein expression was also examined from the normal colon of one healthy, male Sprague Dawley rat. Higher ERβ and lower GPER expression was detected in this normal colon compared to CRC cell lines (Figure 2). HT29 cells were derived from a young female CRC patient, distinguishing them from DLD1, SW620 and T84 cells which are all of male-origin. We therefore selected HT29 cells as the primary model for our further study into the mechanism of GPER activity.

### 2.3. Wnt Overactivation Induces Nuclear Accumulation of β-Catenin Which Is Not Impacted by GPER Agonist G1

To investigate if GPER activation can regulate activity of the Wnt pathway, HT29 cells were treated with the GPER agonist G1 for 24 h and nuclear accumulation of β-catenin was assessed by confocal microscopy. Under basal conditions, β-catenin was concentrated in the plasma membrane. Cultured cells in the presence of G1 (250 nM) for 24 h did not show any detectable changes in levels of nuclear β-catenin (Figure 3). The GSK3β inhibitor CHIR99021 (CHIR) inhibits the activity of the β-catenin destruction complex and thereby activates the Wnt pathway. Incubation of HT29 cells with CHIR (4 µM) for 5 h resulted in a significant translocation of β-catenin to the nucleus (Figure 3). Pre-treatment of these cells for 24 h with G1 followed by a co-treatment with G1 and CHIR for 5 h induced a comparable nuclear translocation of β-catenin to that seen with CHIR alone (Figure 3). Thus, G1 treatment appears not to alter β-catenin nuclear translocation under these conditions.

### 2.4. GPER Agonist G1 Prevents the Increase in JUN Expression Which Is Induced by Wnt Overactivation

Though an impact of G1 on β-catenin nuclear accumulation could not be detected, we felt it was important to investigate if G1 treatment could impact downstream expression of common Wnt target genes. We identified three genes that changed robustly in response to CHIR treatment in HT29 cells: *AXIN2*, *KCNN4* and *JUN*. The GSK3β inhibitor CHIR induced a significant, concentration-dependent increase in *AXIN2* and *JUN* and a decrease in the K^+^ channel *KCNN4* (Figure 4). Treatment of HT29 cells with G1 for 48 h had no significant impact on expression of these 3 genes compared to vehicle control, in the absence of CHIR (Figure 4). To investigate if G1 could impact CHIR-induced Wnt activity, we pre-treated the cells for 24 h with G1 followed by co-treatment with G1 and CHIR. G1 had no significant impact on the upregulation of *AXIN2* or downregulation of *KCNN4* induced by CHIR. However, pre-treatment of these cells with G1 prevented the CHIR-induced increase in *JUN* expression. Treatment with CHIR (4 µM) induced a 2.7-fold increase in *JUN* in the absence of G1 but this increase was reduced to a non-significant 1.1-fold change in response to CHIR when G1 was present (Figure 4). The impact of G1 in the presence of CHIR was highly significant (*p* < 0.01). This identifies a mechanistic role for GPER in modulating the transcriptional output of Wnt signalling in the female-derived HT29 cells.

It has previously been reported that male and female rat colon tissue does not respond in the same way to estrogen exposure [24]. With this in mind, future in vivo or ex vivo experiments are warranted to explore in detail the circumstances under which GPER can impact Wnt signalling. To gain a preliminary insight to this subject, we treated our panel of 4 CRC cell lines (1 female, 3 male) with G1 for 48 h and monitored gene expression of *JUN*. No significant change in *JUN* expression was detected in HT29, DLD1 or T84 cells under basal conditions. A significant decrease in *JUN* was detected in male-derived SW620 cells in response to G1 (Appendix A) which may indicate the capacity for GPER to impact Wnt signalling in male colonic cells also.

### 2.5. GPER Association with Survival Varies by Gender and Tumour Stage

It has previously been reported that high GPER gene expression associates with poor relapse-free survival in females with late stage disease but not in females with early stage disease (Marisa dataset) [17]. No association with survival was detected in males at any stage. Due to the inconsistencies in the literature, we were keen to see if this observation held true in a second, independent dataset. The Guinney dataset does not have information on GPER gene expression and many samples from the SieberSmith dataset are missing information about CRC stage. Thus, in order to assess a dataset of comparable size, we looked at the TCGA Colorectal adenocarcinoma Pancancer dataset using cbioportal (*n* = 579 patients) [25]. Consistent with results from the Marisa dataset, we found from the TCGA dataset that high *GPER1* expression significantly associated with shorter survival in stage III and IV female CRC patients whereas no association was observed at stage I and II (Figure 5). *GPER1* expression did not associate with survival rates in male CRC patients of any stage. Taken together, our results validate the previously published *GPER1* associations of the Marisa cohort for the first time in an independent cohort.

## 3. Discussion

In this study we have identified a sexually dimorphic correlation between gene expression of the Wnt receptor, *FZD1*, and survival in CRC patients. We identify for the first time, a potential protective role for GPER in maintaining low expression levels of the oncogene Jun, in the presence of hyperactivated Wnt signalling in a female-derived CRC cell line. We also validated in an independent patient cohort, a previously reported observation that the association between *GPER1* expression and survival differs depending on gender and tumour stage.

At first glance, these results may appear contradictory: if GPER can play a protective role then it might be expected that high expression of GPER would associate with better survival. However, the Kaplan–Meier analysis is observational and cannot attribute causation. There are many other similar examples such as ERalpha expression in breast cancer which is known to promote tumour proliferation and yet high expression of ERalpha robustly associates with good survival rates. The validation of the gender- and stage-specific survival association of GPER in CRC in an independent data set is an important finding, particularly given the scarcity of reproducible results in this field. Our findings highlight the relevance of GPER to CRC patient survival and the need to consider both gender and tumour stage in future studies as we strive to dissect the functional role of GPER.

The nature of the interaction between GPER activation and Wnt/β-catenin signalling is highly selective. Our original hypothesis was based on previous data reporting anchoring of β-catenin at the plasma membrane by a complex of E-cadherin and the estrogen regulated K^+^ channel KCNQ1 [26]. We anticipated that GPER may contribute to the anchoring of β-catenin and play a global role in regulating Wnt/β-catenin transcriptional activity. However, our data shows no impact of G1 on total β-catenin levels or localisation but rather a specific impact on individual transcriptional targets of this pathway, such as Jun. The molecular nature of this interaction is yet to be determined. It may be that GPER directly regulates specific co-factors which influence some but not all canonical Wnt signalling transcriptional responses. Alternatively, GPER activation may, more indirectly, influence distinct signalling pathways such as the MAPK pathway to influence select consequences of Wnt signalling such as Jun expression. Further studies will be required to tease apart the molecular mechanisms at play. Furthermore, of note, Jun is known to form a complex with β-catenin and TCF4 and to stabilise this transcriptional complex [27]. Thus, it is conceivable that the maintenance of low levels of Jun detected in our study could lead to reduced global Wnt signalling over a longer time course than the single time point examined in our study.

Having identified that GPER can selectively oppose the transcriptional response to Wnt signalling, the next step will be to confirm if this is a protective or detrimental effect. To date studies have suggested both roles for GPER. The most obvious interpretation is that maintaining low levels of the oncogene Jun would be protective, reducing the proliferation of tumorigenic cells. Supporting this, we have previously shown that treatment with the GPER agonist G1 (1 µM) can significantly reduce proliferation and migration in HT29 cells [17]. However, the assumption that Jun is involved in this functional outcome should not be presumed without empirical testing. Moreover, Gilligan and colleagues have previously demonstrated that G1 can increase proliferation in both HT29 and HCT116 CRC cells when given in doses of 1–1000 nM for 72 h [15]. Similarly, a low dose of the non-steroidal estrogenic mycotoxin, Zearalenone, has also been shown to enhance proliferation of HT29 and SW480 cells [28]. However, Liu and colleagues observed that G1 decreased proliferation and increased apoptosis in HCT116 and SW480 cells treated with doses of 600–1000 nM G1 for 48 h [16]. Loss of cell viability has also been demonstrated in SW620 cells in response to 1 µM G1 [29]. In our study, 250 nM G1 was sufficient to restrict Wnt-induced Jun expression. It will be important to establish the functional consequences of this restriction and whether they are impacted by the dose of G1 administered.

In our model CRC HT29 cell line, we used pre-treatment with G1 as well as co-treatment with CHIR to prevent the full impact of Wnt activation. Establishing whether pre-treatment is necessary for this response will be important to determine if G1 also has the capacity to reverse the impact of Wnt activity, or only to prevent the impact from occurring in the first place. Wnt targeting therapies are not currently used in the clinic. The activity of the Wnt pathway is essential for tissue homeostasis and regeneration, hence, blocking the pathway for therapeutic reasons has been shown to cause side-effects such as bone fragility, fatigue, nausea, diarrhoea or alopecia [30]. Thus, an alternative strategy that might suppress Wnt tumorigenic effects without inhibiting the entire pathway would be very attractive. Targeting GPER has been looked at in other clinical settings. Preclinical studies have shown anti-tumorigenic effects of GPER activation in melanoma by suppressing cMyc and PDL1 [31]. This has led to the initiation of early phase clinical trials targeting GPER either as monotherapy or in combination with anti-PD1 antibody pembrolizumab [32]. Understanding the role of GPER at different stages of tumor development in CRC will be important to help us maximise use of such compounds in the future. More immediately, estrogens and estrogen regulating compounds (SERMS) are already widely used in the clinic in a variety of settings. This extends to the testing of dietary phytoestrogens for patients undergoing surveillance colonoscopy [33]. It is imperative that we improve our understanding of sexual dimorphism and the role of estrogen signalling in CRC in order to give appropriate advice on the safe and effective use of such compounds.

In conclusion, our study highlights sexual dimorphism within the Wnt signalling pathway in CRC. We identify a role for GPER in modulating the transcriptional output of Wnt signalling, by restricting CHIR-induced gene expression of the oncogene Jun in HT29 cells. This has the potential to offer protection against CRC progression although the functional consequence is yet to be explored.

## 4. Materials and Methods

### 4.1. Colon Tissue, Cell Culture and Treatments

Colon cancer cell lines HT29, DLD1 and SW620 were cultured in Dulbecco’s modified Eagle’s medium phenol-red free (Gibco, Waltham, MA, USA) at 37 °C in a humidified incubator with 5% CO_2_. T84 cells were maintained in a 1:1 mixture of DMEM and Ham’s F-12 phenol-red free (Sigma, St. Louis, MO, USA) and the breast cancer cell line MCF7 was grown in Eagle’s Minimum Essential Medium (Sigma). All media were supplemented with 10% Foetal Bovine Serum (FBS), 50 u/mL penicillin-streptomycin (Gibco) and 2 mM L-glutamine (Gibco). The culture media was renewed every 2–3 days and cells were sub-cultured once they had reached 70–80% confluence. All experiments were performed within 12 passages. In order to reduce the estrogen content of the culturing media in the experiments, the FBS was stripped with charcoal-dextran (cds, Sigma-Aldrich, St Louis, MO, USA) following manufacturer’s indications. CRC cells were serum-starved for 24 h prior to treatment with 250 nM G1 (Bio-Techne, Minneapolis, MN, USA), 4 µM CHIR99021 (Sigma) or vehicle (DMSO) in 1%cds-FBS for the corresponding time (5 h–48 h). The G1 and CHIR99021 stock solutions were stored at −20 °C in aliquots in order to minimize freeze–thaw cycles. Cell lines were authenticated by STR profiling and routinely tested for mycoplasma contamination. Normal colon was dissected post-mortem from a 14-week old male Sprague Dawley rat. The colon was flushed with ice-cold PBS, cut into small fragments and snap frozen. Tissue was subsequently thawed and minced on ice in RIPA buffer for protein extraction.

### 4.2. Western Blot Analysis

Protein samples were obtained by washing cells with ice-cold PBS, spinning down cell suspensions and resuspending the cell pellet in RIPA buffer (50 mM Tris-HCl pH 7,5, 150 mM NaCl, 1 mM EDTA, 1% Triton X-100, 0.1% SDS) complemented with protease inhibitor cocktail (Roche, Basel, Switzerland) and phosphatase inhibitors (Roche). The DC Protein Assay (Biorad) was used to determine total protein concentration in each sample according to manufacturer’s indications. 30 μg of total protein were submitted to 10% SDS-polyacrylamide gel electrophoresis in a Mini-PROTEAN Electrophoresis System (Bio-rad, Hercules, CA, USA). Proteins were transferred to a PVDF (polyvinylidene fluoride) membrane (Bio-Rad) by a semi-dry transfer at 20 volts for 1 h. Membranes were blocked in 5% BSA (bovine serum albumin) in Tris-Buffer Saline-Tween 20 for 1 h at room temperature and incubated with the pertinent primary antibody overnight at 4 °C: ERalpha (Cell Signaling Technology, Danvers, MA, USA, #8644, 1/1000), ERβ (Cell Signaling Technology, #5513, 1/1000), GPER (Abcam, Cambrisge, UK, #ab154069, 1/500) and β-actin (Sigma, A5316, 1/5000). Following incubation with HRP-conjugated secondary antibody (Cell Signaling Technology) for 1 h at room temperature the chemiluminescent signal was detected using an Amersham Imager 600 (GE Healthcare, Chicago, IL, USA). If necessary, the PVDF membrane was stripped using the Restore Western Blot Stripping Buffer (Thermo, Waltham, MA, USA) for 15 min at room temperature and then probed again using a different primary antibody. Densitometry analysis was conducted using Image J software. Relative protein expression was normalised to β-actin and to HT29 cells (ERα and GPER) or SW620 cells (ERβ) as there was a sample of these cells on each Western blot membrane to be compared.

### 4.3. Real-Time Quantitative PCR (RT-qPCR) Analysis

Total RNA was extracted from cell pellets using the RNeasy Mini Kit (Qiagen, Venlo, The Netherlands) and it was complemented with an in-column DNase digestion to ensure removal of any genomic DNA (Qiagen). First strand cDNA synthesis was performed using 1 μg of total RNA and the ImProm II Reverse Transcription System (Promega, Madison, WI, USA) as per the manufacturer’s instructions. mRNA expression levels of several genes were assessed by quantitative real-time PCR (qPCR) on a 7500 HT Fast System. *ESR1* and *ESR2* expression levels were analysed using a TaqMan PCR master mix (Biosciences) and a specific TaqMan probe for each gene (ESR1, HS01046816_m1; ESR2, Hs01100353_m1). Expression levels of *ACTB*, *GPER1, ERa36*, *AXIN2*, *JUN* and *KCNN4* were determined with Fast SYBR Green PCR master mix (Bioscience) and the following primers: *ACTB* forward 5′-GACGACATGGAGAAAATCTG-3′ and reverse 5′-ATGATCTGGGTCATCTTCTC-3′; *GPER1* forward 5′-ACAAACCCAACCCAAACCAC-3′ and reverse 5′-CACCGTGCAGCTTTCAAGAT-3′; *ERa36* forward 5′-TCTGCAGGGAGAGGAGTTTG-3′ and reverse 5′-TGAGGCCTTATGACCAGAGG-3′; AXIN2 forward 5′-GCCGCATTCAAGTGCAAACT-3′ and reverse 5′-TGCAAAGACATAGCCAGAACC-3′; *JUN* forward 5′-CAAGAACTCGGACCTCCTCA-3′ and reverse 5′-TCCTGCTCATCTGTCACGTT-3′; and *KCNN4* forward 5′-GCTGCTGCGTCTCTACCTG-3′ and reverse 5′-AAGCGGACTTGATTGAGAGCG-3′. Fold change in expression was calculated relative to *ACTB* (housekeeping gene) using the 2^−ΔΔCt^ formula. The experiment with 8 conditions (cells ± G1 ± CHIR at multiple concentrations) was repeated with *n* = 7 biological repeats. Statistical analysis was performed with a two way ANOVA with Tukey’s post hoc test.

### 4.4. Immunofluorescence Analysis

Cells were seeded on 13 mm diameter glass coverslips (Thermo scientific, Waltham, MA, USA) in 12-well plates and incubated with the according treatment solution. Cells were fixed with 4% paraformaldehyde in PBS for 20 min at room temperature. Permeabilization was performed for 10 min with 0.1% Triton X-100 in PBS and subsequently, coverslips were blocked in 0.1% Tween-20, 1% BSA, 22 mg/mL glycine in PBS for 30 mins. Thereafter, the cells were incubated with the β-catenin primary antibody (Millipore, Burlington, MA, USA, #05-665) diluted 1/50 in a glycine-free blocking buffer (0.1% Tween-20, 1% BSA in PBS) overnight at 4 °C in a humidified chamber. The secondary antibody (Alexa Fluor 488-conjugated anti-mouse, Thermo Scientific, Waltham, MA, USA) was applied diluted 1/200 in glycine-free blocking buffer for 2 h in darkness at room temperature in a humidified chamber and the coverslips were mounted in Vectashield (Vector Laboratories, Burlingame, CA, USA) containing 4′,6′-diamidino-2-phenylindole (DAPI) stain. Mounted coverslips were stored at 4 °C in an opaque box for at least 48 h before acquiring the images using a Zeiss LSM710 laser scanning confocal microscope (×60 objective lens). Nuclear β-catenin was quantified with software Image J from 1 μm thick confocal images. β-catenin fluorescent signal was quantified in each nuclear area delimited by DAPI. 4 representative images with a minimum of 10 nuclei were quantified per condition in each experiment to obtain the average nuclear β-catenin expression. The experiment was repeated with 4 biological replicates and statistical analysis of nuclear β-catenin levels was performed using the repeated measures one-way ANOVA with Tukey’s post hoc test.

### 4.5. Kaplan–Meier Analysis

Survival associations of the expression levels of *FZD1* were evaluated in male and female CRC patient populations by Kaplan–Meier survival analysis. 4 publicly available datasets were assessed using the R2 bioinformatics web tool (R2: Genomics analysis and visualization platform http://r2.amc.nl (accessed on 23 May 2022)): Guinney, Marisa (GSE39582), SieberSmith (GSE14333 + GSE17538) and Beissbarth (GSE87211). Patients were divided into high or low expression levels of *FZD1* based on the median gene expression. The default probe-set in R2 bioinformatics web tool (the verified probe-set expressed by the largest number of samples) was used. *p*-values were calculated based on a log rank test. Additionally, a Kaplan–Meier analysis of *GPER1* expression levels was conducted in male and female cohorts of The Cancer Genome Atlas (TCGA) dataset using the cBioPortal for Cancer Genomics site (https://www.cbioportal.org/ (accessed on 8 February 2022)) [25,34,35]. A sub-stratification based on TNM stage was also included and patients were divided into early stage (TNM I and II) versus late stage (TNM III and IV). Given the low expression levels of *GPER1* in the TCGA cohort, patients were divided by the last quartile (Q4, highest expression).

## Figures and Tables

**Figure 1 ijms-23-12581-f001:**
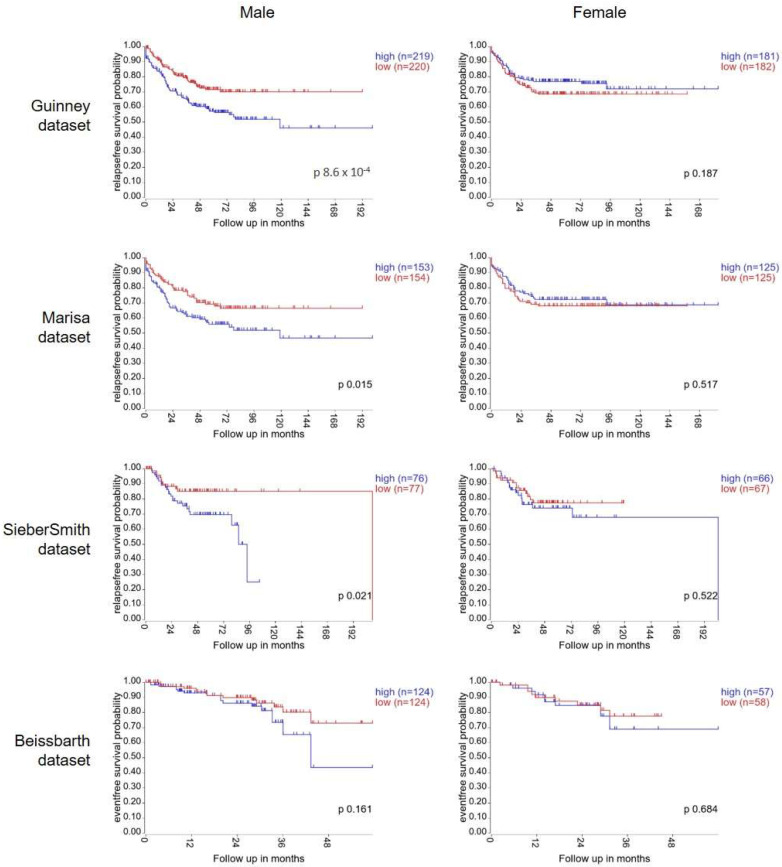
High gene expression of Wnt Receptor, *FZD1*, associates with poor survival in male CRC patients but not in female CRC patients. Kaplan–Meier curves showing relapse-free survival rates (Guinney, Marisa and SieberSmith datasets) or event-free survival rates (Beissbarth dataset) for colorectal cancer patients with up to 200 months of follow up. Data from male patients are plotted on the left-hand side with female patients on the right-hand side. Patients were stratified into high and low expression of the Wnt receptor *FZD1*, based on the median expression. *p* values were calculated with a log-rank test by the R2 bioinformatics webtool (http://r2.amc.nl (accessed on 23 May 2022)).

**Figure 2 ijms-23-12581-f002:**
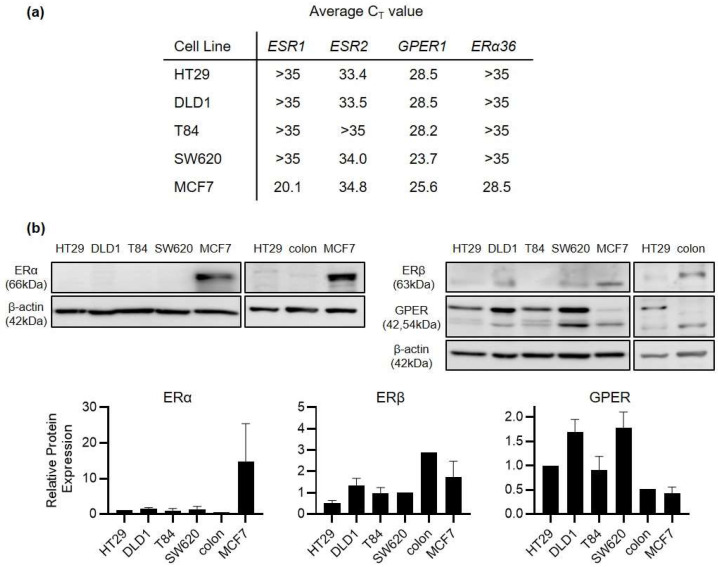
GPER is the predominant estrogen receptor expressed in CRC cell lines. RNA and protein expression of estrogen receptors was compared across HT29, DLD1, T84 and SW620 CRC cell lines as well as the MCF7 breast cancer cell line. (**a**) Mean C_T_ values are shown for 4 estrogen receptors (ESR1, ESR2, GPER1 and ERα36) across 5 different cell lines as measured by real-time PCR. The lower the C_T_ value, the higher the relative expression of that gene. A C_T_ value > 35 indicates a gene which is expressed below the level of reliable detection. *n* = 3 replicates from each cell line. (**b**) Western blot analysis of protein expression of ERα, ERβ and GPER across the same 5 cell lines. Images are representative of *n* = 3 replicate blots (*n* = 4 for GPER). For comparison, receptor expression is also compared between HT29 cells and the normal colon tissue from one male Sprague Dawley rat. Bar graphs show densitometry analysis of relative receptor protein expression normalised to beta actin (mean ± SEM).

**Figure 3 ijms-23-12581-f003:**
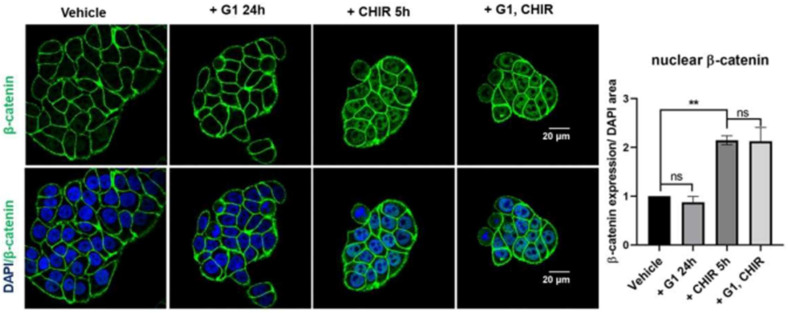
Wnt-induced β-catenin nuclear translocation is not altered by pre-treatment with G1 in HT29 cells. Confocal microscopy images and corresponding nuclear β-catenin quantification of HT29 cells pre-treated with 0.25 µM G1 for 24 h and subsequently co-treated with 0.25 µM G1 and 4 µM CHIR for 5 h in 1%cds-FBS. Green is β-catenin staining, blue is nuclear DAPI staining. Nuclear β-catenin was significantly increased after 5 h of CHIR incubation in HT29 cells. G1 treatment alone did not significantly impact nuclear β-catenin levels in HT29 cells. Co-treatment with G1 did not prevent CHIR-induced accumulation of nuclear β-catenin, *n* = 4. Statistical analysis was performed using the repeated measures one-way ANOVA with Tukey’s post hoc test. Mean ± SEM; ** *p* ≤ 0.01; ns, not significant.

**Figure 4 ijms-23-12581-f004:**
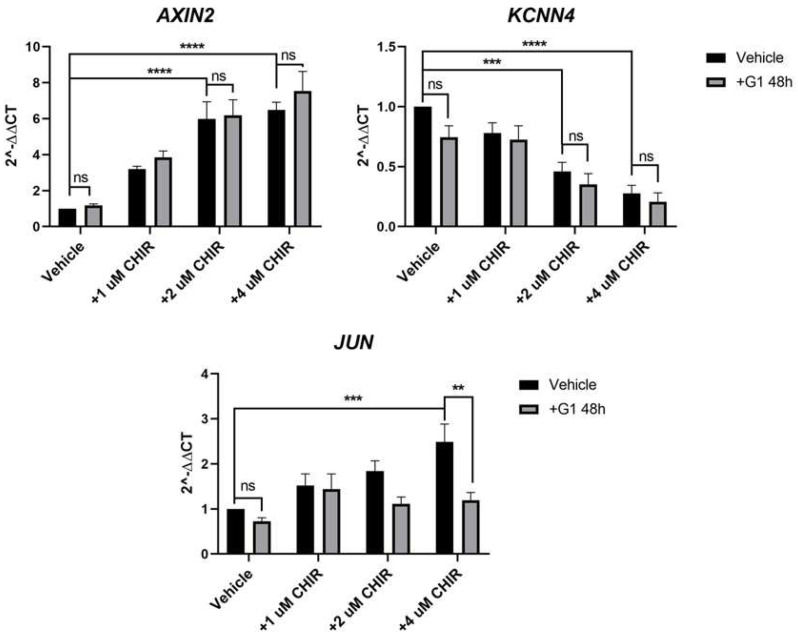
Pre-treatment with G1 prevents CHIR-induced upregulation of *JUN* in HT29 cells but does not alter CHIR-induced regulation of *AXIN2* or *KCNN4*. HT29 cells were pre-treated with 0.25 µM G1 for 24 h and subsequently co-treated with 0.25 µM G1 and 0−4 µM CHIR for a further 24 h in 1%cds-FBS containing media. RT-qPCR analysis shows CHIR itself increased *AXIN2* and *JUN* expression levels whereas it reduced *KCNN4* expression levels in a concentration dependant manner in HT29 cells, reaching the highest significance at the highest CHIR concentrations. Co-treatment of HT29 cells with G1 significantly prevented CHIR-induced increase in *JUN* mRNA levels. However, G1 did not impact CHIR-induced change in *AXIN2* nor *KCNN4* expression levels, *n* = 7. Statistical analysis was performed with a two-way ANOVA analysis with Tukey’s post hoc test. Results from the 2-way ANOVA analysis showed *AXIN2* expression levels were significantly impacted by CHIR (*p* < 0.0001) but not G1 treatment (*p* = 0.2647); in the case of the expression levels of *KCNN4* both CHIR (*p* < 0.0001) and G1 treatment (*p* = 0.0392) were found to exert a significant impact overall; more notably *JUN* expression levels were shown to be modulated significantly by both CHIR (*p* = 0.0004) and G1 treatment (*p* = 0.0005). Mean ± SEM; ** *p* ≤ 0.01, *** *p* ≤ 0.001, **** *p* ≤ 0.0001, ns, not significant.

**Figure 5 ijms-23-12581-f005:**
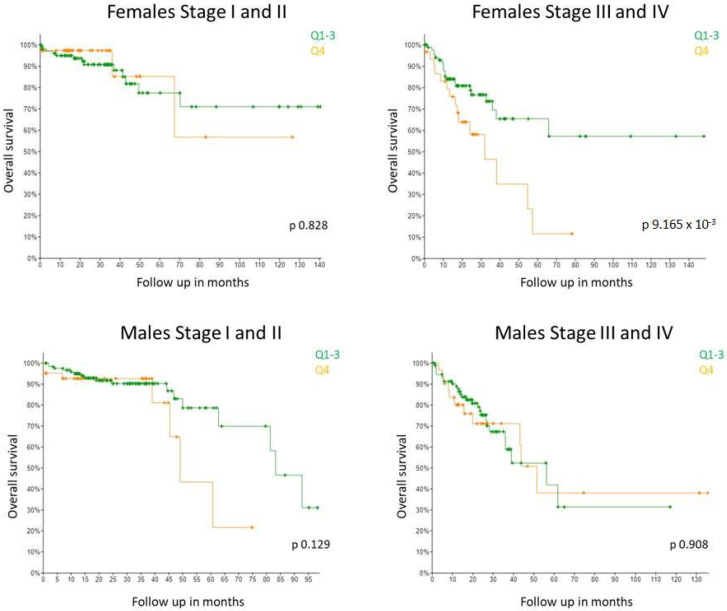
High *GPER1* expression significantly associated with worse overall survival rates exclusively in TNM stage III and IV CRC female patients, and did not correlate with survival in TNM stage I and II nor male patients at any stage in the TCGA dataset. Kaplan–Meier survival analysis by *GPER1* expression in TCGA cohort divided by the last quartile (Q4 vs. Q1–3). Increased *GPER1* expression (Q4) significantly correlated with poor survival in the 150 females at stage III and IV (*p* = 9.165 × 10^−3^) and not in the 130 females at stage I and II (*p* = 0.828). In the male CRC patients, *GPER1* expression did not associate with survival at any stage (168 at stage I and II; 131 at stage III and IV) (*p* = 0.129; *p* = 0.908).

## Data Availability

Not applicable.

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
