# Peer review of "GPER Agonist G1 Prevents Wnt-Induced JUN Upregulation in HT29 Colorectal Cancer Cells"

_ijms, 2022, doi:10.3390/ijms232012581_

Round 1

Reviewer 1 Report

The manuscript by McBryan and group highlights sexual dimorphism within the Wnt signalling 306 pathways in CRC. The study is well planned and executed to get the results which are supported by proper explanations wherever necessary. They identified role for GPER in modulating the transcriptional output of 307 Wnt signalling. Overall, the manuscript is well written, neatly presented and of interest to IJMS readers. Therefore, I suggest to accept the manuscript for publication in IJMS in its current form.

Author Response

We thank Reviewer 1 for taking the time to review our manuscript in detail and providing such a favourable review. We are grateful for the suggestion to accept the manuscript for publication in its current form.

Reviewer 2 Report

This is an interesting study examining the potential role of the novel estrogen receptor GPER/GPER1 in mediating sex selective protective effects upon colorectal carcinoma. While the study is generally performed well, there are some concerns regarding the presence of various estrogen receptors that should be examined in greater detail, as presented below:

1. In figure 2, the western blots for estrogen receptors are of generally poor quality. Please use advanced quantification software to determine the density values of the protein bands, normalize these values to their corresponding beta-actin levels, and present such data in graphical form.

2. In figure 2, protein expression data on the 3 different ERs from 'normal' colorectal epithelium must be presented as a control for the CRC cells used.

3. Given the poor quality of western blot images, it is strongly recommended to use an alternative modality such as immunohistochemistry or immunofluorescence, to demonstrate the protein expression (or the lack of it) of the 2 ERs in CRC, MCF-7, and normal colorectal epithelial cells. In addition, such data would also demonstrate the respective localization of the 3 ER types in these cells. As a matter of fact, ERalpha and ERbeta, while typically considered as nuclear, have also been shown on cell membranes, while GPER can be localized within the cell nucleus (Chakrabarti and Davidge Plos One2012;7(12):e52357)

4. Finally, the novel estrogen receptor GPER is also known as GPER1 and GPR30. it is strongly recommended to use these alternate terminology, both in the Abstract and Introduction

Author Response

We are grateful to Reviewer 2 for taking the time to review our manuscript in detail and providing such a favourable review. In response to these suggestions, we have now improved Figure 2 to include an additional “normal” colorectal control. Responses to each individual point raised are provided below:

  1. As suggested, quantification software (Image J) has now been used to quantify the protein expression for each of the estrogen receptors in each of the cell lines. Values have been normalised to the corresponding beta-actin levels and densitometry values are presented in graphical form as part of Revised Figure 2.
  2. As a control, protein from the colon of a male Sprague Dawley rat was isolated and compared with CRC cells for expression of estrogen receptors. As has previously been reported, the normal rat colon expressed more ERbeta and less GPER than the CRC cell lines. This data has been added to Figure 2.
  3. We thank the author for the suggestion to conduct immunofluorescence, however, the subcellular localisation of the estrogen receptors is beyond the scope of this manuscript. Findings from the western blot analysis are in agreement with those previously reported in the literature.
  4. We are grateful for the suggestion to acknowledge the alternative names for GPER. This information has now been added to the text of the Abstract (line 15) and Introduction (line 55).

Reviewer 3 Report

This is an exciting topic that shows estrogenic signals could help in the prevention of colorectal cancer. In previous studies, they have demonstrated how GPER activation can inhibit proliferation, mitochondrial-related apoptosis, and cell arrest. Targeting GPER could be a potential therapy for the treatment of colorectal cancer.

In this article, the authors have analyzed that Wnt-receptor FZD1 is associated with poor survival rates in the male but not the female population. In addition, GPER agonist G1 treatment does not affect β-catenin nuclear translocation in female-derived HT29 CRC cell lines. It would be interesting to know if β-catenin nuclear translocation is affected by G1 treatment in male-derived cell lines as these cell lines have a higher protein level of GPER. Also, the author can analyze the expression of Wnt target genes in male-derived cell lines as data suggests the correlation between the Wnt pathway and poor survival.

The author showed that upregulation of JUN could be prevented by treatment with GPER agonist G1 in HT29 cells. To confirm these findings, the author can analyze how overexpression of GPER in HT29 cells affects the activation of the Wnt pathway. In addition, over-expression of GPER or G1 treatment in male-derived cell lines could reveal its role in Wnt activation by analyzing β-catenin nuclear translocation and expression of their target genes. It will show whether male or female-derived cell lines have similar effects of G1 agonist or not.

Author Response

We are grateful to Reviewer 3 for taking the time to review our manuscript in detail, for appreciating how exciting the topic is, and for providing valuable suggestions for further study.

As recommended, new data has now been added to the Results section, which compares the G1 response in male- and female- derived CRC cell lines. We are wary of over interpreting this data due to the small and unbalanced number of female and male cell lines commercially available and the fact that there are other differences aside from gender origin between the cell lines. Nevertheless, results do indicate that GPER activity may be able to modify expression of the Wnt target gene Jun in male-origin cells as well as female-origin cells and we acknowledge in the text that this warrants further investigation. This data is presented in the Results section lines 188-196 and in Supplementary Figure S1.

We appreciate the suggestion to conduct GPER overexpression experiments and had previously considered this. However, given the already abundant nature of GPER expression in these cell lines (Ct values 23 to 29), such experiments would appear futile and are unlikely to add meaningful results to the study.

Round 2

Reviewer 2 Report

No further comments.